# LMD3: Language Model Data Density Dependence

**John Kirchenbauer**[*1], **Garrett Honke**[*2]
**Gowthami Somepalli**[1], **Jonas Geiping**[3,5], **Daphne Ippolito**[4,6], **Katherine Lee**[6]
**Tom Goldstein**[1], **David Andre**[2]
[1]University of Maryland, [2]X, the Moonshot Factory
[3]ELLIS Institute Tübingen, [4]Carnegie Mellon University
[5]MPI-IS, Tübingen AI Center, [6]Google DeepMind

## Abstract

We develop a methodology for analyzing language model task performance at the individual example level based on training data density estimation. Experiments with paraphrasing as a controlled intervention on finetuning data demonstrate that increasing the support in the training distribution for specific test queries results in a measurable increase in density, which is also a significant predictor of the performance increase caused by the intervention. Experiments with pretraining data demonstrate that we can explain a significant fraction of the variance in model perplexity via density measurements. We conclude that our framework can provide statistical evidence of the dependence of a target model's predictions on subsets of its training data, and can more generally be used to characterize the support (or lack thereof) in the training data for a given test task.

## 1 Introduction

*"With the right dataset and the right benchmark, you can make a deep learning model fake absolutely any ability. It's easiest if the benchmark and the dataset are the same thing, but otherwise you can also achieve it with a dataset that is a dense sampling of the distribution the benchmark is sampled from." - @fchollet*

The goal of this study is to perform a careful series of experiments designed to provide concrete evidence for, or against, the hypothesis that the accuracy of a model on a test question is strongly impacted by the density of the training set around that question. To do this, we explicitly estimate a probability density function on the training distribution. We then ask whether it is possible to explain a significant amount of a language model's abilities by directly estimating how *dense* the training data distribution is at each test point. We quantify the density of training data in embedding space using a classic technique in statistics, *Kernel Density Estimation* (KDE), which relies on measurements of the similarity between a test point and its neighbors. We present an illustration of the general process in Figure 1.

To determine whether such a simple technique offers *any* explanatory power in the realm of modern large language models, and to assess the impact of density on test performance, we begin by manipulating density through contamination of a training dataset with copies (or near copies) of test samples. We then observe the correlation between accuracy on test samples and the training density near those samples. We confirm that 1) a properly configured kernel density estimate can detect the increased density near these training points and 2) the elevated performance is reliably predicted by the density estimate value for the question texts.

We then study the natural density variation within a pretraining corpus and uncover a more complex and subtle downstream effect. Higher density test points have lower perplexity ("as expected") when density is estimated using only the nearest neighbors in the pretraining corpus for each query, although the effect size is small. However, when estimating density using a random subset of training points, we observe the opposite correlation.

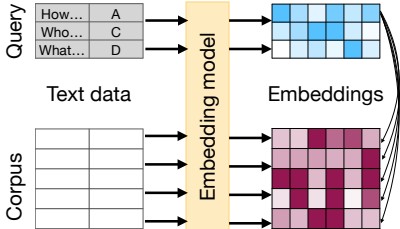 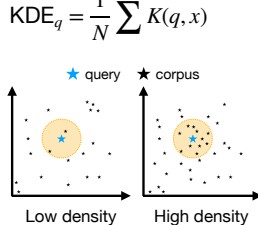

Figure 1: A system level view of the LMD3 pipeline. The corpus of data used to train a LLM and a test set of queries are projected into a vector space using a neural embedding model. For each resulting query vector, a density estimate with respect to the training corpus is computed. The resulting density estimates can be used to infer the model's ability to respond to a question-like query or simply reproduce the tokens in the query sequence based on whether the relative density is higher or lower at that point in sample space.

We interpret our findings within the context of a wide array of prior work on data attribution, data contamination, and the counter-intuitiveness of high-dimensional statistics, and conclude that simple measures of dataset density are indeed sharp enough tools to be predictive of per-sample test set performance. At the same time, we observe that—at least at the 7B scale where models do not dramatically interpolate their data—the effect of dataset leakage can be surprisingly small. We wrap up with a discussion of applications of our methodology to instance and group-wise error analysis, dataset filtering and supplementing interventions, and the determination of test contamination from the training corpus.

## 2 Related Work

### 2.1 Memorization and Contamination

Large language models have been shown to memorize sequences from their training data. The key, replicated finding across this literature is that the more a sample is repeated in the training data the more likely it is to be memorized (Carlini et al., 2022; Hernandez et al., 2022). Repetition is an important extremal case of elevated density corresponding to a spike in the density function at a specific point in the sample space and as such we expect a density based analysis to highlight the causal relationship between repetition and elevated sampling likelihood. Additional work has shown that a similar relationship holds for fuzzy or *near-duplicated* samples (Ippolito et al., 2022) implying that a density measure might also capture this weaker dependence relation.

Recent work has studied the impact of data contamination during the pretraining and finetuning processes. Despite the fact that training solely and repeatedly on verbatim or paraphrased test samples leads to significantly elevated performance (Yang et al., 2023), the impact can be much less pronounced in realistic scenarios where leaked samples are mixed into a larger corpus during pretraining (Jiang et al., 2024). Since leakage of test queries constitutes a practically relevant instantiation of the aforementioned edge case of spikes in the training distribution, we design our initial set of experiments around a leakage intervention.

### 2.2 The "Data Attribution Hypothesis"

Stated informally, the "data attribution hypothesis" is the belief that specific machine learning model behaviors can be attributed to a specific set of examples from the training data. The critical implicit assumption is that the set of training points the behaviors are attributable to is *small* with respect to the overall size of the training corpus. Therefore, work on data attribution seeks to design methods for identifying the set of most relevant examples, with notable approaches including the use of optimization related measures (gradients) to develop "influence functions" (Koh & Liang, 2017), behavioral approximations of model outputs through the lens of kernels (Park et al., 2023), and even exhaustive models of

how a model behaves with respect to leave-one-out resamples of its training dataset (Ilyas et al., 2022). To benchmark these techniques, curated datasets of queries and corresponding fact sets to which correct responses could/should be attributed have also been developed (Akyürek et al., 2022). In this work, we take an orthogonal approach by both relaxing the attribution hypothesis' assertion that the relevant set of training data for each query must be small, and switching away from relying on characteristics of the final model and instead directly analyzing the training corpus itself.

## 2.3 The "Similarity Hypothesis"

Quite related to the data attribution hypothesis is the "similarity hypothesis"—the near truism that models will make predictions that are similar to their training data. While prior work has shown that structured features of training data such as the elevated occurrence of specific named entities correlates with performance on factual test questions concerning those entities (Kandpal et al., 2023), similar results involving more general notions of train-test interrelatedness are fewer and far between (at least in the modern LLM literature). That said, recent work such as Tirumala et al. (2023) has shown that density quantification can be used to guide dataset pruning routines for improved sample complexity during model training. Also published contemporaneously to our research activities, Yauney et al. (2023) showed that a wide variety of similarity metrics were "not enough to explain performance across benchmarks". While certain aspects of our results do suggest that the story is subtle, rather than simple, in contrast to their rather definitive conclusions, we posit that significant variance in LLM performance *can* be explained given the right measure and lens of analysis.

# 3 Preliminaries: Kernel Density Estimation

Density estimation is the general problem of estimating a probability density function from data. In our setting, we aim to estimate a distribution that is intractable, namely the function $P : \mathbb{R}^d \to \mathbb{R}$, where for our domain we have represented all of the points $x \in X$ from our training data as $d$-dimensional vectors, and we would like to approximate the likelihood of drawing any given sample from the distribution over natural language sequences represented by our corpus. The tool we choose to use for this approximation is the Kernel Density Estimate.

**Definition 3.1.** Kernel Density Estimate (KDE) For a training corpus $X_c = \{x_0, x_1, ...x_{n-1}\} \in \mathbb{R}^d$ with a bandwidth parameter $h > 0$ and a kernel function $K_h : \mathbb{R}^d \times \mathbb{R}^d \to \mathbb{R}$, for a query vector $x_q$ the KDE at $x_q$ over $X_c$ denoted $\text{KDE}_{X_c}(x_q)$ is given as:

$$\text{KDE}_{X_c}(x_q) = \frac{1}{|X_c|} \sum_{x \in X_c} K_h(x, x_q)$$

However, because the sum is over all $n$ training samples, the cost of computing the KDE for realistic language model training datasets is intractable. To apply KDE in this setting, we turn to DEANN - Density Estimation from Approximate Nearest Neighbors, a clever optimization proposed and analyzed by Karppa et al. (2022) that enables the scaling of KDE to large datasets $X_c$ by decomposing the full KDE into the exact contributions of close neighbors, and approximate contributions from the rest of the corpus. A formal definition of the approximation algorithm that we use when computing KDEs on pretraining-scale datasets is provided in Algorithm 1. We discuss the details of the approximation scheme further in Appendix A.1.

# 4 The LMD3 Methodology

For a chosen data point (text sequence) we want to study how the dataset density at that point is related to model performance at that point. We measure performance on "task-like" text sequences using model accuracy, and on unstructured or "webtext-like" data using the loss function. Our method takes as input a model and its training data, as well as one

or more test/query sets of interest. The training data are embedded using an embedding model, and the nearest neighbors for each query sequence are retrieved from the vectorized corpus. Then, either, the neighbors are combined with a random subset of corpus vectors to compute a density estimate at each query point, or, when scale permits, the density estimate is computed exactly at each query point.

## 4.1 Computing Embeddings

We take for granted access to performant, pretrained neural sequence embedding models as general purpose similarity/distance functions for pairs of text segments. We consider two types of embedding spaces in which to perform density estimates, the features produced by an off-the-shelf *retrieval* model and the features derived from the hidden states of the model under analysis. While in preliminary investigations we explored working with the latter "self-derived" features, we failed to find a significant difference between the two in our small scale (fine-tuning) experiments, and so we stick with the former for large-scale experiments. The use of LLM hidden states as retrieval features is an active area of study and future versions of our methodology could benefit from those results, so we leave a more comprehensive evaluation of exactly what self-derived LLM embeddings represent that general purpose retrieval models do not to future work.[1]

## 4.2 Computing the KDEs

We utilize the software package developed by the DEANN authors (Karppa et al., 2022) as an extremely efficient parallelized implementation of the exact KDE calculation for euclidean kernels when the data $(X_q, X_c)$ fits in memory. Since bandwidth selection is an empirical process, we perform an initial evaluation across a range of bandwidths in preparation for our paraphrase experiments (Appendix A.10).

For the finetuning-scale experiments, where the corpus contains $\sim 100,000$ samples, we are able to directly use the exact version of the KDE. For experiments with a pretraining dataset, we leverage the authors' exact KDE implementation as well as their random sampling based KDE approximation as subroutines. Algorithm 1 describes how we combine this with a scalable neighbor search system to implement our "fully decomposed" version of the nearest neighbor search-based approximate KDE. "Fully decomposed" simply means that before running the KDE calculations themselves, as a preprocessing step for the query set, the neighbors for each example are retrieved from the training corpus via a massively distributed vector search engine (*exactly*, in our case, as opposed to *approximately* as suggested by DEANN).[2] Then, the random component for each density calculation is selected in a hierarchical manner via an initial large random sample without replacement, and a second sampling step, also without replacement, to select the final random complement for each query's individual density estimate. In Section 5.3 and Section 6.1, "KDE" refers to the exact KDE computed over the entire finetuning corpus, while in Section 5.4 and Section 6.2 it refers to the final value yielded by Algorithm 1—the weighted average of the local and random components of the KDE approximation over the pretraining corpus, except where denoted specifically as solely the "local" or "random" component.

---

[1]We also remark that running multi-billion parameter embedding models over billions of sequences is costly and therefore such models are not as amenable to the pretraining scale experiments we ultimately perform.

[2]While the details of the neighbor search system we use in this work for the *pretraining* scale experiments are proprietary, we include the finetuning-scale implementation in our code. The neighbor search system can be substituted with any vector search engine that scales to the data under analysis, even if that scale necessitates an approximation scheme such those implemented in the FAISS library (Johnson et al., 2019). Karppa et al. (2022) argue that this additional approximation does not bias the overall KDE approximation results any further.

---

**Algorithm 1** Fully Decomposed Approx. KDE with Hierarchical Sampling

---

**Input:** A corpus $X_c$ of $n$ text embeddings $x_c \in \mathbb{R}^d$, a $k$ nearest neighbor search subroutine over vectors in $X_c$, $NN_k(\cdot)$, a kernel function $K$ and bandwidth parameter $h$ together with the corpus over which it is computed $X$, defining a $KDE_X(\cdot)$, two random sample size parameters $m_1$ and $m_2$ ($m_2 < m_1 \ll n$), a dataset of query embeddings $X_q, x_q \in \mathbb{R}^d$.

**Output:** $Z_q, \in \mathbb{R}_{>0}^{|X_q|}$, an approximation of the KDE for each $x_q \in X_q$.

Randomly sample w/o replacement $X_1$ of size $m_1$ from $X_c$
**for all** $x_q \in X_q$ **do**
   $X_{nn} \leftarrow NN_k(x_q) \quad \in \mathbb{R}^{k \times d}$
   $X_1' = \{x \in X_1 | x \notin X_{nn}\}$
   Randomly sample w/o replacement $X_2$ of size $m_2$ from $X_1'$.
   $z_{nn} \leftarrow KDE_{X_{nn}}(x_q)$
   $z_{rand} \leftarrow KDE_{X_2}(x_q)$
   $z_q \leftarrow (\frac{k}{n})z_{nn} + (\frac{n-k}{n})z_{rand}$
**end for**
{Note that we can return $Z_{nn}$ and $Z_{rand}$ individually to analyze the effect of local and global contributions independently.}

---

## 5 Experiments

### 5.1 Models and Data

For our experiments we use a transformer-based sequence embedding model from the `sentence-transformers` library (Reimers & Gurevych, 2019) to generate the feature space in which we compute density estimates. For the controlled finetuning experiments we train the base Llama 2 7B model (Touvron et al., 2023) on doctored versions of the training set for the MMLU benchmark for 2 epochs as this regime produces meaningful differences between experimental settings whilst still being realistic (and resulting in a model that has not completely degenerated in other regards). For the pretraining experiments we analyze the performance of the 6.9B model from the Pythia suite (Biderman et al., 2023) as a function of a version of its training corpus, The Deduplicated Pile (Biderman et al., 2022). We provide further details on these choices and experimental parameters in Appendix A.2.

### 5.2 Paraphrasing Process

In order to develop our density methodology, we design a synthetic experimental setting in which we can directly tune the specific amount of support for test queries present in the training data. Roughly defined, an idealized *paraphrase* of a test example is the transform from natural language sequence $x$ to $x'$ such that the semantics of $x$ and $x'$ are equivalent. Access to such a transformation function allows us to perform controlled experiments where we take a test sample $x$ and intervene to increase the amount of specifically relevant training data for this sample $x$ by mixing paraphrases $x'$ into the training data. This can be done in a number of ways, and we discuss the design choices for this process further in Appendix A.3.

### 5.3 Controlled Experiment 1: Leakage to Increase Density, Finetuning Scale

In our first series of experiments, we take $1,000$ random questions from the MMLU test set of $14,042$ questions and paraphrase them 3 times. We also consider an exact copy of the selected test questions as a "paraphrase" with perfect similarity, or distance 0 to the original query. For each test question we paraphrase, we compute the cosine similarity between the original query and each sample in the set of 3 paraphrases based on their embeddings. Considering the paraphrases for each query sorted in *ascending order by similarity* to the original query as Para 1, Para 2, and Para 3 respectively, we then refer to the collection of

two paraphrases {Para 1, Para 2} as Paras = 1,2 and the collection of all three paraphrases {Para 1, Para 2, Para 3, } as Paras = 1,2,3.

Then we finetune our base language model on datasets constructed by taking the "auxiliary train" split of the MMLU dataset and mixing in various combinations of the paraphrases and/or an exact copy of the original query, which we denote the presence of, or lack thereof, using Exact = 1 and Exact = 0 respectively. We evaluate the performance of the models trained on these different datasets and analyze the impact that the inclusion of paraphrases and exact copies of test queries has on the trained model. "Rank Accuracy" denotes the scoring method used by the HuggingFace Open LLM Leaderboard for evaluating MMLU, first proposed by Brown et al. (2020).

## 5.4 In-the-Wild: In and Out-of-Distribution Queries, Pretraining Scale

Finally, we turn our focus to the loftier goal of relating the test time behavior of a language model to its *pretraining* data. In addition to the increased scale with which we apply the technique, the increased generality of the pretraining distribution requires us to curate sets of queries to represent different test time scenarios we might be concerned with. We consider two classes of query sets: in-distribution (ID) and out-of-distribution (OoD), where for the former we choose the simple, explicit definition of any query $x_q$ taken directly from the training corpus $X_c$ as ID, and consider anything else to be OoD. The training corpus is The Deduplicated Pile, re-segmented into $\sim 3.5B$ individual text sequences. For these experiments we compute the approximate KDE using parameters $k = 1,000$ nearest neighbors for the local component, $m_1 = 1,000,000$ samples as a base random sample set, with $m_2 = 10,000$ samples per query selected as the random complement, exclusive to the k nearest neighbors.

**Random 10k segments** (ID): We randomly sample, without replacement, $10,000$ segments from the training corpus itself. These are drawn from precisely the same set of segments over which neighbor retrieval and KDE computation are performed. Since these are simply webtext segments, there is no notion of "response" so we just compute perplexity (PPL) of the text segment under the LLM.

**MMLU Test** (OoD): We employ the same $14,042$ test questions as in the finetuning experiments to act as a set of queries that are not contained in the training data. Since there are ground truth responses for the queries, we can compute perplexities on both the query texts as well as on the correct target response when conditioned on the query text.

**OpenOrca Random 10k** (OoD): As another set of diverse sequences outside of the train distribution, we consider a random selection of $10,000$ questions from the OpenOrca curation project's main dataset. Since these also have reference responses, we can compute perplexities on both the query and responses for these samples.

## 6 Results

Considering the relative novelty of our proposed methodology and in order to de-risk the main experiments, we performed a series of validation analyses regarding the embedding model used for the paraphrasing experiments, the bandwidth hyperparameter of the KDE, and basic series of finetuning runs to calibrate our expectations about exactly how much a model should overfit when trained repeatedly on verbatim leaks of test questions. We discuss these preliminary experiments in section Appendix A.10. For the main experiments we consider bandwidths of $\{0.01, 0.05, 0.1, 1.0\}$ for the exponential kernel, and $\{0.1, 0.2, 0.5, 1.0\}$ for the gaussian kernel though we only report a small selection across the main body figures, primarily gaussian at 0.1 as a "narrow" setting tuned for discrimination in the leakage experiments and 0.5 as a "wide" setting for capturing a broader range of similarities in the pretraining-scale experiments.

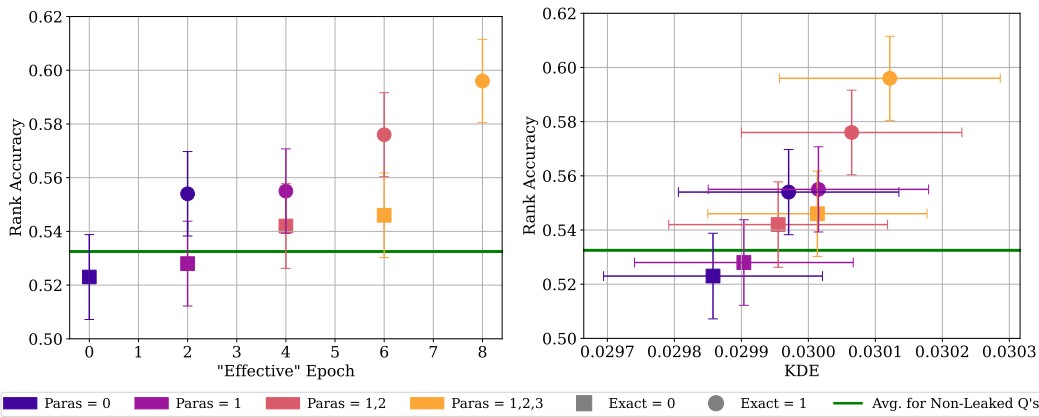

Figure 2: **Left)** To enable the aggregate interpretation of the paraphrasing experiments, we plot accuracy on leaked test questions as a function of the number of "effective epochs" the model has trained on the leaked questions for. **Right)** We plot the same performance measure as a function of KDE, with gaussian kernel and bandwidth 0.5. We see that the trend in accuracy according to our density measure corresponds with the trend in accuracy according to the known degree of leakage, effective epochs.

## 6.1 Controlled Experiments: "Expected" Dependence between Performance and Density.

In Figure 2 we demonstrate that training on the various leaky dataset formulations improves performance by visualizing changes in performance across different leakage settings. We enable this by first plotting performance against a handcrafted feature called "effective epochs", where we compute the number of epochs the model has effectively trained on a given test question $x_t$ based on the specific set of paraphrases and exact duplicates $X_p$ included in the training set. For simplicity, in this figure, we represent exact copies and paraphrases equally such that if we train for 2 epochs, and have leaked 1 exact copy and 1 paraphrase, the total number of times the test question is trained on would be $2 * (1 + 1) = 4$. In Figure 2 we observe both a positive trend in performance as a function of effective epochs as well as a positive trend as a function of our KDE measure—the latter of which is computed *without* access to either ground truths or any outputs of the LLM being analyzed.

In Figure 3, we present accuracies and densities across the full factorial manipulation of exact and paraphrased leaks. The figure shows that the KDE is a discriminative feature between the leak set and non-leak set. To measure the reliability of the correspondence between data density and performance, we perform mixed-effects regressions and discuss these results in detail in Appendix A.5. In summary, we find that the leakage conditions reliably increase accuracy (Exact leak: $p < .001$, Paraphrased leaks: $p < .001$) and decrease perplexity (Exact leak: $p < .001$, Paraphrased leaks: $p < .001$) on the test questions. Critically, training data density estimates also reliably predict variance in accuracy and perplexity where as density increases, accuracy increases ($p < .001$) and perplexity decreases ($p < .001$).

## 6.2 In-the-Wild Experiments: Aggregation-Specific Dependence Relationship.

First considering a set of random samples from The Deduplicated Pile itself, ID with respect to the model's training data, in Figure 4 we plot perplexity as a function of KDE by binning the data by KDE values (x-axis) and computing the average accuracy (y-axis) within each bin. In the left chart we see that when measuring the KDE with respect to only the random samples selected in our approximation algorithm we get a noisy, slightly positive trend (which is surprising[3]). However, when we consider the KDE computed only with respect to the nearest neighbors in the corpus, we get a sharp negative trend, in line with results in the

---

[3]...but potentially explainable by distances in high dimensions being counterintuitive. Being closer on average to a collection of random other points effectively means being close to nothing in particular.

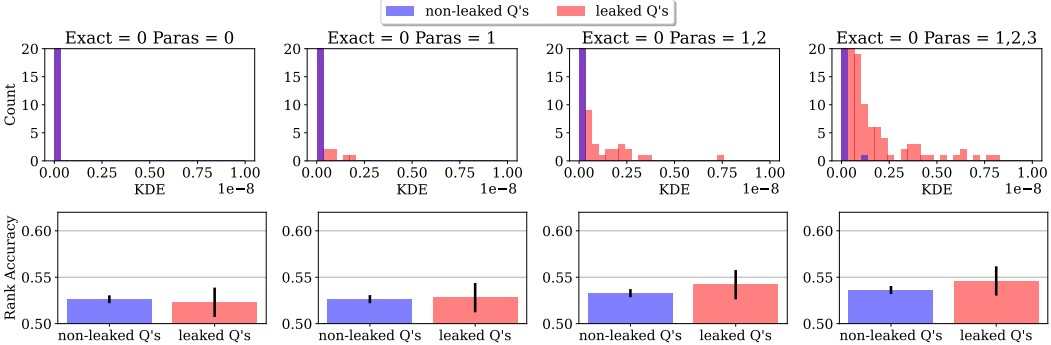

(a) Experiments where no exact copies of test questions are included, only paraphrases. For "Count" histograms, the y-axes are cropped to $[0, 20]$ and x-axes to $[0, 10^{-8}]$ to showcase differences between non-leaked and leaked. An uncropped version is provided in Figure 7.

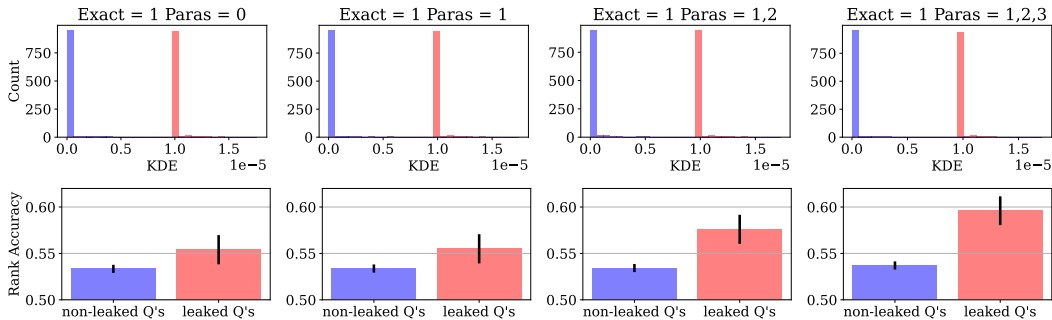

(b) Experiments where 1 exact copy of test questions are included, in addition to paraphrases. Note that axes for "Count" histograms are uncropped in this panel.

Figure 3: Moving from left to right in **(a)** and **(b)** shows the effect of an increase in the number of paraphrases of each test question that are leaked into the training data. Experiments in **(b)** also include an exact copy of each leaked test question, while experiments in **(a)** do not. **"Count" histograms)** We plot the distributions of KDE values (gaussian kernel and bandwidth 0.1) for the test queries that were leaked, exactly and or via paraphrase, and not leaked, for each leakage intervention experiment. **Accuracy bar charts)** We show the corresponding accuracy breakdown for the leaked and non leaked sets for each experiment. Overall, we find that increasing support for test questions via incorporating paraphrases into the training data increases performance on those test questions, and this increase is magnified by the addition of exact leaks of test questions. The addition of the exact copy of each question also makes the leaked and non-leaked question sets highly *separable* under our KDE measure as demonstrated by the distinct concentration of "leaked Q" KDE values away from 0.0 in **(b)**.

finetuning setting.[4] This is affirmed by the trend relating each query's average distance to its top $k$ neighbors to model perplexity on said query. (*Note* that x-axes are reversed in the distance charts to make the trend visually congruent with the handedness of the trends for density measurements Figure 4 and Figure 5.)

Next, switching to a set of OoD queries, the MMLU test set, the left chart of Figure 5 shows that the PPL on the query/question texts is not strongly correlated with the local KDE component, but the middle chart shows that it *is* correlated with the average distance to the top k neighbors in the corpus. Further, we see that the perplexity of the correct response is also correlated with distance to the top k neighbors.

---

[4]We report this random - local breakdown explicitly here because at pretraining scale, the coefficient used to combine the local and random components according to Algorithm 1, $k/N$ versus $(N-k)/N$ respectively, causes the random component completely wash out the local contribution.

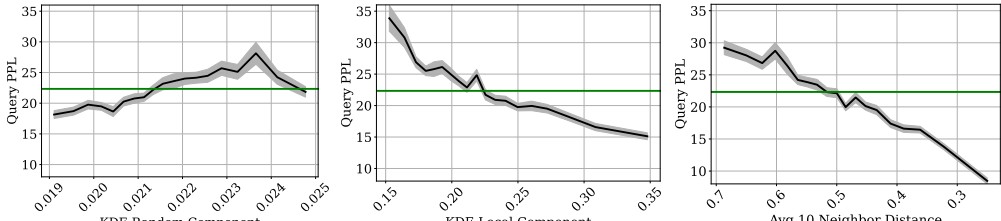

Figure 4: Perplexity according to Pythia 6.9B for random samples from The Deduplicated Pile (ID) as a function of KDE with gaussian kernel and a bandwidth of 0.5, marginalized via equal mass binning into 20 bins. **Left)** Query perplexity vs. the KDE with respect to a random sample of points in the corpus. **Middle)** Query perplexity vs. the KDE with respect to only the local neighborhood within the corpus. **Right)** Query perplexity vs. the average distance to the to the top k neighbors. Horizontal line denotes the average across all queries. We see that while the trend in Query PPL as a function of the random component of the KDE is non-monotonic, and even weakly positive, when considering the local region of highly similar samples for each query, there is a strong clear negative trend in PPL as a function of density, as measured by the local KDE or a simple average over neighbor distances.

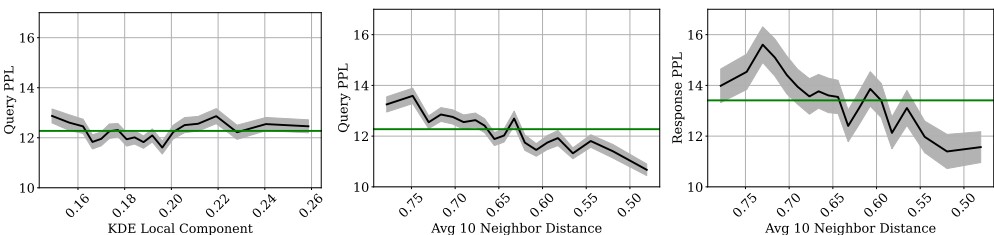

Figure 5: Perplexity according to Pythia 6.9B for questions from the MMLU test set (OoD) as a function of KDE with gaussian kernel and a bandwidth of 0.5 or average distance to k nearest neighbors, marginalized via equal mass binning into 20 bins. **Left)** Question perplexity vs the KDE with respect to only the local neighborhood within the corpus. **Middle)** Question perplexity vs distance to k nearest neighbors. **Right)** *Response* perplexity vs distance to k nearest neighbors. Horizontal line denotes the average across all queries. While the relationship between query perplexity and local KDE isn't particularly strong, there is a stronger trend as a function of simple neighbor distances. For response perplexity, we see a clear trend as a function of average neighbor distances.

As with the finetuning scale experiments, we used mixed-effects regression to measure the reliability of the effect of density on perplexity and report those full results in Appendix A.9. In summary, we find that *query* perplexity decreases as data density increases for the randomly-sampled ID query set ($p < .001$) and the OoD OpenOrca dataset ($p < .001$) but not the OoD MMLU Test query set ($p = .95$). However, *response* perplexity does decrease slightly with increased density for the MMLU Test set ($p < .001$).

Since the text length directly effects perplexity values, for Figure 4 and Figure 5 we must manually isolate a subset of the queries where lengths are relatively similar to remove the variance due to length before plotting. We also observe a small number of extremely large outlier perplexity values and drop those rows as well. Details about this process are provided in Appendix A.7. We report only the most insightful combinations here and present a more exhaustive series in Appendix A.8.

# 7 Insights and Applications

We can summarize our key insights from both groups of experiments succinctly by stating that in extremal cases like test question leakage, one can expect a clear relationship between model performance and apparent training data density in query space. However, at pre-training scale, the specific way in which one aggregates information about relevant support

in the dataset matters. Using the abstraction of "effective epochs" one can understand how repeated training on subsets of the data can lead to elevated marginal performance on those subsets but we caution that the number of effective epochs required to achieve drastically inflated test accuracies on benchmarks, such as those reported in Yang et al. (2023), might be much higher than one would expect. This suggests that any un-evidenced claim that contamination is a reason to *wholly* discount the benchmark results for models trained on unspecified data distributions (at least at the common 7B parameter scale) should be taken with a grain of salt.

Overall we believe that training data density quantification is a *grounded* methodology for analyzing the failure modes of language models at an instance and group-wise level, since it builds evidence for expected relative performance based directly on aspects of *the training data itself*. In response to observing relatively high density in certain regions of test query space and lower density in others, we expect that the consistency of model performance could be improved by supplementing data in those regions of weaker support through human data curation as well as automated processes such as machine paraphrasing.

## 8 Limitations and Conclusion

A few key limitations of our study include the focus on a small set of specific, but very relevant, datasets and models for the field and the fact that the KDE based methodology we propose is not hyperparameter free—the user must choose the kernel parameters. That said, the KDE methodology stands apart from simpler aggregations like the top-$k$ distance measure in its "smoothness". While our experiments show that avg. distance to the top-$k$ neighbors is also a reliable predictor of variance in performance, that metric requires the choice of a specific value of $k$. Neighbors that happen to lie beyond this boundary will not be accounted for *at all* in a simple top-$k$ aggregation and thus changing $k$ by just a value of 1, can alter the measurement significantly. In contrast, the KDE measure accounts for the nearest and farthest neighbors simultaneously and varies smoothly under small perturbations of the bandwidth hyperparameter as well as minor changes to which points fall in or outside of the $k$ nearest neighbors set.

It is also likely that the preprocessing parameters—length and stride—used to segment the training data before embedding are critical to the outcome of the analysis. We work at a finer segmentation granularity than Tirumala et al. (2023) and Yauney et al. (2023) but we do not ablate our choices enough to prove that this is required to explain meaningful amounts of variance. Finally we also acknowledge that full access to the training dataset of a newly "released" model is increasingly rare. However, given the potential of data-centric analysis techniques such as ours we claim that this is a shortcoming of contemporary norms within the field rather than a true limitation of our methodology. Rather, we implore the community to insist on the release of details sufficient to exactly re-materialize the full training data distribution as a necessary precondition for presenting model releases as bona fide scientific artifacts.

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

# A Appendix

## A.1 Approximate Kernel Density Estimation

We refer the reader to the excellent work of Karppa et al. (2022) for complete derivations but describe here the critical results that we rely on in this study. Their Lemma 3 states that one can compute an unbiased estimator of the true $KDE_{X_c}(x_q)$ by splitting $X_c$ of size $n$ into two non-overlapping subsets, $X_A$ and $X_B$, computing $z_A = KDE_{X_A}(x_q)$ and $z_B = KDE_{X_B}(x_q)$ independently, and then combining them in a weighted sum according to the sizes of $X_A$ and $X_B$:

$$z_q = \frac{|X_A|}{n} z_A + \frac{|X_b|}{n} z_B$$

Lemma 2 in their work, itself a result from prior literature, states that we can also compute an unbiased estimator of $KDE_{X_c}(x_q)$ by taking a random sample $X_m$ of size $m$ from $X_c$ and achieve an approximation $KDE_{X_m}(x_q)$ with bounded error that decreases as $m$ increases.

Together, the authors use these results to propose Density Estimation from Approximate Nearest Neighbors (DEANN), where they approximate $KDE_{X_c}(x_q)$ by computing the contribution of the nearest neighbors ($X_A$) and the contribution of the rest of the data ($X_B$). We leverage their results to allow us to perform density estimations at scale across millions or billions of data points. An implementation stylized description is provided as Algorithm 1.

### A.1.1 On the Definition of the Sample Set $X_c$

While the text documents that comprise a training corpus (which we embed to produce $X_c$) are in some sense taken as an *input* to the analysis we propose, the notion of how $D_c$ is broken up into the set of text segments $D_c = \{s_0, s_1, ... s_{n-1}\}$ is actually a design choice both during the training of a LLM and during our density analysis. A simple limitation that must be considered is that the embedding models used to transform $s_i$ into $x_i$ often have a maximum input length they can accept. More critically however, the kernel function that a KDE is based on relies on the ability to compute *meaningful* distances between points in the embedding space.

Choosing the *segment* size for partitioning the documents in $D_c$ defines the sample space over which densities are computed and therefore potentially the outcome of the entire analysis. It is simple to convince yourself of this by considering extreme choices like *all* of the training tokens being a single segment, or every individual token being its own segment (i.e. devolving to a unigram distribution).

Since a comprehensive search over all segmentations is impossible, and we are unaware of conclusive results on optimal segmentation for general retrieval applications let alone our text sequence density estimation problem, for our finetuning experiments we avoid segmenting as the queries are already relatively short, and for the pretraining corpus we analyze, we adopt a simple choice in this work as described in Appendix A.2.5. In Section 8 in the main body we briefly discuss the effect that this choice may have on the results.

### A.1.2 "In Distribution" Queries

An important detail when analyzing queries that are suspected or known to be "In Distribution" (ID), i.e. contained in the training data, is to ensure to embed all samples into the space in which the KDE is performed under the "same conditions". This means that if $x \in X_q$ and $x \in X_c$ then the embedding vector of the query should be *equal* to the embedding of the matching doc in corpus, up to some floating point error tolerance. If any segmentation and/or prompt templating is used, it must be applied in the same manner between the train corpus and test query set if any of the queries are intended to be perfect matches with elements in the corpus with a distance $\sim 0.0$ between their embeddings. For our leakage experiments and for our ID pretraining experiments, we ensure that this requirement is met.

### A.2    Models, Data, Hyperparameters

### A.2.1    SentenceTransformers

For all the finetuning scale experiments involving the MMLU dataset, we utilize the `multi-qa-MiniLM-L6-cos-v1` model from the `sentence-transformers` library (Reimers & Gurevych, 2019) following the setup of Yang et al. (2023). We choose this model based on prior experience, literature precedent, generally favorable placement on the `sentence-transformers` retrieval model leaderboard, and critically it's extreme efficiency. For the pretraining data embeddings, we use a slightly more general purpose variant of the same architecture, `all-MiniLM-L6-v2`, a similar size model trained on a more general data distribution.

We remark that while there are many larger, more competitive retrieval embedding models available, our desired to embed at the billion sample scale limits our choices. We feel that this tradeoff is compensated for by the fact that, at least for the pretraining-scale experiments, we consider very short sequences - just 50 whitespace separated tokens or 3 to 4 English sentences. We rely on the intuition that less representational complexity is required to accurately compare and contrast sequences of this length than when considering much longer documents. We additionally note that most prior work incurs the information loss cost of truncation at lengths of $\sim 512$ since fine-grained re-segmentation is seldom performed and most open source retrieval models have context widths shorter than the multiple thousands now commonplace for more modern auto-regressive models.

### A.2.2    Llama 2

We adopt a focused approach and consider one of the standard state of the art open source large language models, Llama 2 (Touvron et al., 2023), in its base form as the starting checkpoint for our finetuning scale experiments. We perform full finetuning of all parameters using the `axolotl`[5] finetuning harness with the following features: DeepSpeed ZeRO-2, "target only loss" calculation, sample-packing, warm-up with a linear learning rate schedule, and weight decay. We detail the precise parameters for the different experiments in the Appendix A, as well as the training configuration files provided with the source code release.

We also explore computing embeddings based on the last layer hidden states of the Llama 2 model after training on the finetuning data. These features are pooled and normalized in the same manner as the last layer hidden states of the `sentence-transformers` embedding models. In initial investigations we explored the "last layer, last token" strategy of extracting embeddings, which has been explored in contemporary work including Tirumala et al. (2023), but did not find compelling evidence that this more expensive embedding process yields features more useful for our density estimation methodology than features from a smaller retriever model.

We that said, we admit the choice to restrict ourselves to the general purpose, extremely efficient retrieval models is mostly a practical one enabling pretraining scale experimentation and consider the study of how to optimally leverage the internal representations of autoregressive large language models as important area of research (Wang et al., 2023).

### A.2.3    MMLU

We use the MMLU dataset (Hendrycks et al., 2021b;a) as our test-bed for the density estimation methodology. We focus on this dataset because it features prominently in both the Hugging Face Open LLM Leaderboard task set, as well as recent work competing for state of the art results (Team et al., 2023) in language model training.

In order to limit confounding factors, we train the base model on a data distribution similar to that of the test task: the "Auxilliary" training set for MMLU. We format the training samples using the same logic that the EleutherAI LM Evaluation Harness (Gao et al., 2021)

---

[5]`github.com/OpenAccess-AI-Collective/axolotl`

implements. We use the precise MMLU test set as filtered and prepared by the harness, noting that this is a slight subset of the original test set released by Hendrycks et al. (2021b).

### A.2.4 Pythia

The Pythia suite (Biderman et al., 2023) is an open science focused language model collection with public code and *data*. Given that our methodology requires access to the training set, we have limited options for pre-training scale analyses. While none of the Pythia models are directly comparable to Llama 2 due to architectural, training and dataset differences, since it is a relatively modern causal language model it is the most similar option amenable to our analysis at the time of this study.[6]

### A.2.5 The Deduplicated Pile

The Pile (Gao et al., 2020) is a public multi-domain web corpus for language model pre-training. We use the reworked version that was globally deduplicated (Biderman et al., 2022) before the corresponding set of Pythia models were trained on it. The raw dataset itself contains approximately 134M documents, of widely varying lengths. Due to the input length limitations of embedding models, and more importantly to ensure that distances between relatively short queries and individual samples in the corpus will be meaningful, we re-segment the dataset by first splitting the documents into chunks of 50 whitespace separated tokens (split only on " ") with a stride of 40 tokens, creating an overlap of 10 tokens each with preceding and succeeding segments. The result is an inflation of the original 134M documents into $\sim$ 3.5B segments.

[**start**]We make this somewhat arbitrary choice for segment length and stride based on the observation that 50 whitespace separated tokens is approximately the length of a few sentences or a short paragraph in English. The current and previous sentences in this paragraph comprise precisely 50 such "words" on their own.[**end**]

We chose our stride in order to limit false negatives during nearest neighbor search without causing the segment count to grow beyond what the implementation can handle[7]. While we are unable to ablate this choice due to compute limitations, without obvious prior work to compare to, we accept these uncertainties as part of the cost of exploration.

### A.2.6 Approximate KDE Parameters

For each approximated KDE computation in the experiments with The Deduplicated Pile, for each query, we use the exact $1,000$ nearest neighbors, and a random complement of $10,000$, drawn for each query from a fixed pre-sample of $1,000,000$ sequences from the training corpus. These correspond to $k$, $m_2$ and $m_1$ respectively in Algorithm 1.

### A.3 Paraphrase Process Details

In this work, as in Yang et al. (2023) and other work that utilizes paraphrasing as a black-box transformation in the security and safety domain (Kirchenbauer et al., 2023; Krishna et al., 2023), we prompt a powerful large language model, GPT4-turbo, to behave as a general purpose paraphrasing model using a template specifically designed to elicit paraphrased samples that are thoroughly transformed with respect to the original queries. Figure 6 shows the the prompt we use to paraphrase the test queries as well as example of an original MMLU test question and its paraphrase.

Yang et al. (2023) explores adversarially regenerating paraphrases until a paraphrase with sufficient dissimilarity to the original example is produced (in order to evade leakage

---

[6]The Tiny-llama model was trained concurrently to this research, however, the dataset used to train it is still significantly larger than the deduplicated Pile and harder to materialize on disk exactly as it was prepared during the training of that model.

[7]A stride greater than or equal to your segment size results in no overlap between corpus segments, or worse, omitted sequences which could incur "false negatives" during neighbor search even for queries with large overlaps with training data.

```
Please rephrase the following question without
altering its meaning, ensuring you adjust the
word order appropriately. Ensure that no more
than five consecutive words are repeated and
try to use similar words as substitutes where
possible. Do not change the format of the
multiple-choice question. When encountering
mathematical formulas, please try to
substitute the variable names. Ensure the
formulas aren't identical to the original.
When you come across a single number or
letter, consider replacing it with a sentence.
When encountering a long sequence of numbers,
if they are separated by spaces, you can
replace the spaces with commas; if separated
by commas, you can replace them with spaces.
Consider the prompt and choices as a whole;
there shouldn't be consecutive words. If
options are challenging to rephrase, consider
altering the initial letter's
case.\n\nOriginal question:\n\n<QUERY>"
```

ORIGINAL

The following are multiple choice questions (with answers) about sociology.

Marxist feminists explain patriarchy in terms of:

A. a lack of equal rights and opportunities for men and women

B. sex classes, through which men oppress women economically, politically and sexually

C. women's domestic labour being exploited by the capitalist economy

D. the dual systems of capitalism and male domination

Answer:

PARAPHRASED

Below are several multiple-choice inquiries concerning sociology, each provided with answer selections.

How do Marxist feminists interpret the concept of patriarchy?

A. An absence of equivalent rights and opportunities between the genders

B. Gender strata, whereby females are subjugated by males on economic, political, and sexual grounds

C. The exploitation of female household work by the system of capitalism

D. The twofold structures of capitalist society and masculine hegemony

Answer:

Figure 6: The prompt used to instruct the powerful LLM used as a paraphrasing engine for the MMLU experiments, adopted from Yang et al. (2023). Similar to their work, we find that GPT-4 Turbo, `gpt-4-1106-preview` version, is able to reliably paraphrase MMLU questions without losing too much context. We visually point out to the reader the fact that the entire question preamble as well as all answer choices are part of the paraphrasing step.

detection measures). In our work, we are instead interested in collecting a spectrum of paraphrases with varying degrees of similarity and dissimilarity to test queries, and therefore, we simply sample $k$ paraphrases for every test query on which we intervene, finding that pairing the prompt with the `gpt-4-1106-preview` model yields diverse paraphrases that are still faithful to the key details of the original question.

## A.4 Regression Analyses: Mixed Effects Modeling

In this work, we are interested in measuring the effect of training data density at the example level. We treat example training data density as a *fixed effect* and other covariates (such as the subject area of an MMLU question) as *random effects*—noise we wish to remove. *Mixed effects* modeling works by marginalizing the covariates and fitting a model to predict the dependent variable from the fixed effect(s). The model yields an estimated coefficient for each fixed effect and $p$-values that, used in conjunction with an $\alpha$-criterion, determine the statistical significance of the effect. We conclude the fixed effect reliably predicts the variance of the dependent variable when the $p$-value is below the $\alpha$-criterion, customarily $\alpha = .05$. Concretely, if we determined $p = .001$, we'd expect to mistakenly conclude there was an effect 1 time in 1000 hypothetical experiments.

Importantly, the sign of an estimated coefficient indicates the direction of the relationship. So a fixed effect (e.g., density) would have a statistically-reliable positive correlation with a dependent variable (e.g., accuracy) if $p < 0.5$ and $\hat{\beta} > 0$. Overall, we look for agreement between the observed trends in binned plots of performance marginalized over various ranges of density estimates within a test set, the significance of the fixed effect, and the sign of its regression coefficient.

All analyses were performed using R Statistical Software (v4.3.3, R Core Team, 2021) and supporting packages `lme4` v1.1-35.2 (Bates et al., 2015) and `lmerTest` v3.1-3 (Kuznetsova et al., 2017).

## A.5 Controlled Experiments: Leakage Full Regression Results

We present two classes of analysis for the controlled experiments: manipulation verification regressions and the critical experimental regressions evaluating the effect of training data density. Regressions targeting the DV of rank accuracy are generalized linear mixed models (GLMER) fit by maximum likelihood with the Laplace approximation. Regressions targeting the DV of perplexity are linear mixed models (LMER) fit with restricted maximum likelihood (REML) and we use Satterwaite's method for calculating $p$-values. The mixed effects

structure across all regressions is the same: we treat query length and example nested in MMLU question domain as random intercepts.

### A.5.1 Leak manipulation verification.

**Predicting rank accuracy with leak condition (GLMER)**

$rank\_accuracy \sim exact\_leak + paraphrase\_leaks + (1|len(d_q)) + (1|task(d_q)/example)$

*Note: Due to the natural ordering of paraphrase exposure (0,1,2,3), we treat the paraphrase variable as an ordered categorical and test its relationship as linear, quadratic, and cubic.*

|                              | Estimate | Std. Error | $z$ value | $p$-value |
|------------------------------|----------|------------|-----------|-----------|
| (Intercept)                  | 1.30910  | 0.35018    | 3.738     | $<.001$   |
| exact_leak                   | 0.62556  | 0.07626    | 8.203     | $<.001$   |
| paraphrase_leaks (Linear)    | 0.44949  | 0.07448    | 6.035     | $<.001$   |
| paraphrase_leaks (Quadratic) | 0.08411  | 0.07471    | 1.126     | 0.26      |
| paraphrase_leaks (Cubic)     | -0.08231 | 0.07569    | -1.087    | 0.27      |

**Predicting perplexity with leak condition (LMER)**

$perplexity \sim exact\_leak + paraphrase\_leaks + (1|len(d_q)) + (1|task(d_q)/example)$

|                              | Estimate   | Std. Error | $t$ value | $p$-value |
|------------------------------|------------|------------|-----------|-----------|
| (Intercept)                  | 1.803      | 3.199e-02  | 56.361    | $<.001$   |
| exact_leak                   | -1.642e-01 | 9.318e-03  | -17.620   | $<.001$   |
| paraphrase_leaks (Linear)    | -1.094e-01 | 9.254e-03  | -11.827   | $<.001$   |
| paraphrase_leaks (Quadratic) | 4.924e-02  | 9.318e-03  | 5.284     | $<.001$   |
| paraphrase_leaks (Cubic)     | -1.513e-02 | 9.381e-03  | -1.613    | 0.107     |

### A.5.2 Training data density effect evaluation.

**Predicting rank accuracy with training data density (GLMER)**

$rank\_accuracy \sim KDE_{K,h=0.1} + (1|len(d_q)) + (1|task(d_q)/example)$

|                  | Estimate  | Std. Error | $z$ value  | $p$-value |
|------------------|-----------|------------|------------|-----------|
| (Intercept)      | 1.086     | 3.437e-01  | 3.159      | 0.00158   |
| $KDE_{K,h=0.1}$  | 5.678e+04 | 3.514      | 16159.318  | $<.001$   |

**Predicting perplexity with training data density (LMER)**

$perplexity \sim KDE_{K,h=0.1} + (1|len(d_q)) + (1|task(d_q)/example)$

|                  | Estimate   | Std. Error | $t$ value | $p$-value |
|------------------|------------|------------|-----------|-----------|
| (Intercept)      | 1.90       | 3.129e-02  | 60.70     | $<.001$   |
| $KDE_{K,h=0.1}$  | -2.158e+04 | 9.025e+02  | -23.91    | $<.001$   |

### A.5.3 Unique Variance Explained by KDE versus Top-k Distance

For completeness, we also consider an alternate modeling setup, one where we directly ask whether the two features, density and top-k distance, explain *different* aspects of variance? This question can be studied by running a slightly different regression model than those above that asks whether the two features explain significant parts of the variance in the performance outcome, when controlling for the variance explainable by the other. Our predictors ("Fixed effects") are $KDE_{K,h=0.1}$ and $avg\_top10\_dist$ and target is again *rank_accuracy*.

**Predicting rank accuracy with training data density and avg. top-k distance (LMER)**

$$rank\_accuracy \sim KDE_{K,h=0.1} + avg\_top10\_dist + (1|len(d_q)) + (1|task(d_q)/example)$$

|                 | Estimate   | Std. Error | $z$ value | $p$-value |
|-----------------|------------|------------|-----------|-----------|
| (Intercept)     | 1.416      | 3.514e-01  | 4.029     | $<.001$   |
| $KDE_{K,h=0.1}$ | 4.625e+04  | 5.372      | 8609.363  | $<.001$   |
| $avg\_top10\_dist$ | -3.595e+04 | 8.711e-02  | -4.127    | $<.001$   |

Interpreting the coefficient estimates, we observe that as density goes up, rank accuracy goes up (+ Estimate) and as neighbor distance goes up rank accuracy goes down (− Estimate) as expected and the model suggests these relationships are accounting for *unique variance*. The fact that both p-values are small indicates that both terms remain reliable predictors of variance when included in the same model.

This evidence clearly shows that despite the close relationship between top-k distances and the kernel density measure, in practice, they can each be used to explain different parts of model behavior. Studying this relationship further is an important area for future work. Overall, we believe this regression outcome helps highlight both the distinction between and compatibility of our proposed methodology and the top-k distance metric.

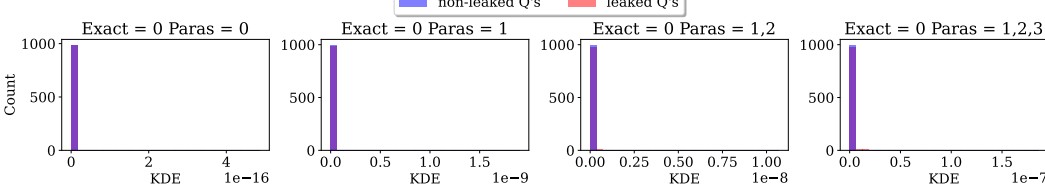

Figure 7: Companion figure to Figure 3a showing the version of panel **(a)** without cropping for the histograms. While the leak and non-leak distributions are indeed distinct both upon inspection and statistical test via regression, this is not visually apparent unless axes are suitably cropped. See main body version.

### A.6 Controlled Experiments: Leave-One-Subject-Out

In a second experiment set focused on controlling the level of training support for specific queries, we utilize the subject metadata provided for the questions in the MMLU testing set to intervene by "leaving-one-out" of the subjects areas covered in the training data. In particular, we consider the *supercategories* defined by the MMLU authors, which maps each of the 57 fine-grained subject areas to a more general topic. The training data does not have subject metadata associated with it and so we generate our own using an automated procedure. However, the results of this experiment are inconclusive.

We assign each one of the training samples from the MMLU "auxiliary" training set to fine-grained subject area using a kNN classifier, where each point receives its label according to a majority vote between the subject labels of the k-nearest questions in the test set according to distances in embedding space. After assigning each training question a subject, we assign

it to a supercategory based on the aforementioned mapping. Since this does not yield a balanced subsetting of the training samples (some supercategories are much larger than others) we examine the counts for each supercategory and select 4 with counts between 2,000 to 4,000 questions (out of 99,842 total).

Then we take the base language model and train it a collection of datasets where for each *split* we leave out the group of training examples corresponding to each of the 4 supercategories selected in turn. For each model we measure the impact that the intervention has on the average performance across test questions sharing the supercategory label of the left out samples, as well as the test questions from the other 3 supercategories as reference. We present the results of these experiments in Figure 8 as deltas in density and performance against the control model, which is trained on the full collection of training samples.

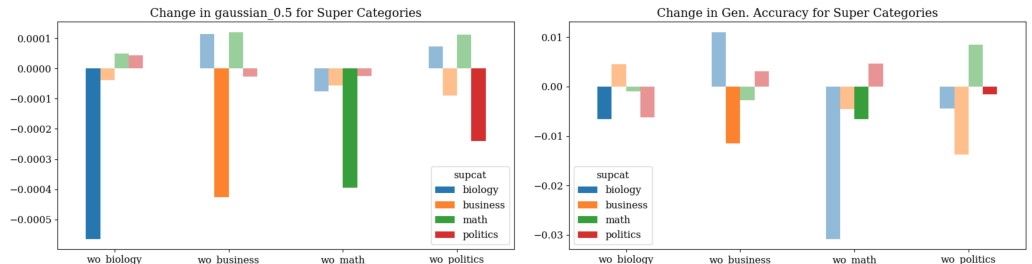

Figure 8: **(Left)** Shows that while the density measurements behave as expected across splits in our leave-one-subject-out experiment, **(Right)** the performance changes are small and inconsistent. Curiously, leaving out the math subset seems to have an outsize impact on the performance of the model on the biology set, suggesting that determining the influence that specific types of training questions have on the model's performance in this setting could be quite difficult no matter which heuristic was chosen.

## A.7   In-the-Wild Experiments: Controlling for length

While the pretraining dataset analyzed, The Deduplicated Pile, is segemented according to whitespace before embedding, there is still a significant amount of variation in the lengths of the texts after tokenization, or as measured by character length. Since perplexity generally decreases as a function of length under any language model as tokens are easier to predict given more context, we need to control for length in order to highlight any observable differences according to density.

We also need to drop outliers with respect to the performance measurement (we drop rows where query PPL > 500 and Response PPL > 60), as these heavily skew the marginalization process. To clean up the data for Figure 4 and Figure 5 in the main body, we first analyze perplexity as a function of query length in characters, and use this view to identify a suitable upper and lower bound for length such that we are able to focus in on any variance that can be explained by changes in density rather than length.

## A.8   In-the-Wild Experiments: Extended Figure Set

While we highlight the most informative figures in the main body (Figure 4, Figure 5) we also present a more complete set of visualizations for the MMLU Test set and Open Orca sampled query set here.

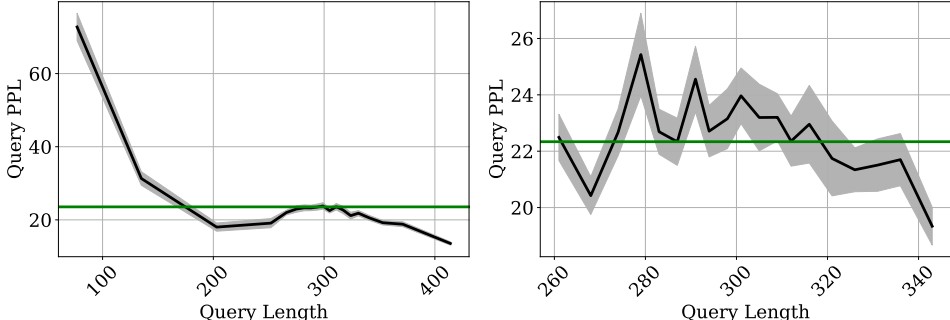

Figure 9: Due to the fact that the length of a text significantly impacts perplexity measurements, **(left)** we analyze the distribution of lengths present in the query set to identify a range of sequence lengths for which the performance measurement is relatively stable. **(right)** For the set of Random 10k (ID) samples from The Pile, we identify this as between 250 and 350 characters, and within this range we see that the query PPL is relatively constant around the mean. Thus, the data is limited to this range for the figures presented in the main body to highlight whatever variance can be attributed to differences in density.

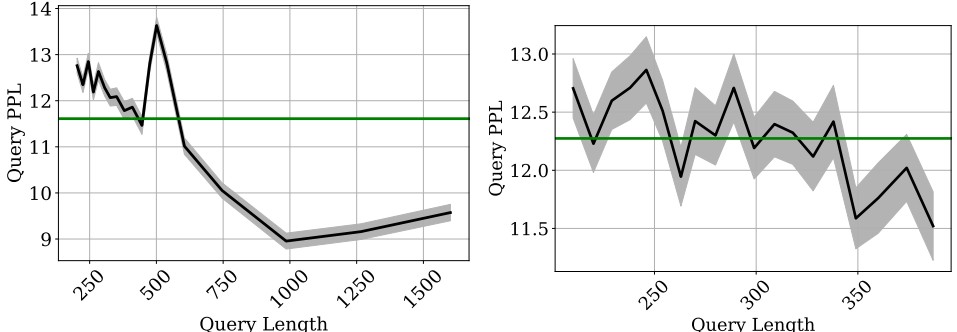

Figure 10: Due to the fact that the length of a text significantly impacts perplexity measurements, **(left)** we analyze the distribution of lengths present in the query set to identify a range of sequence lengths for which the performance measurement is relatively stable. **(right)** For the set of MMLU Test questions, we identify this as between 200 and 400 characters, where the query PPL varies less wildly and where most of the data are concentrated (the majority of the bins are located in the upper left of the left panel). Thus, the data is limited to this range for the figures presented in the main body to highlight whatever variance can be attributed to differences in density.

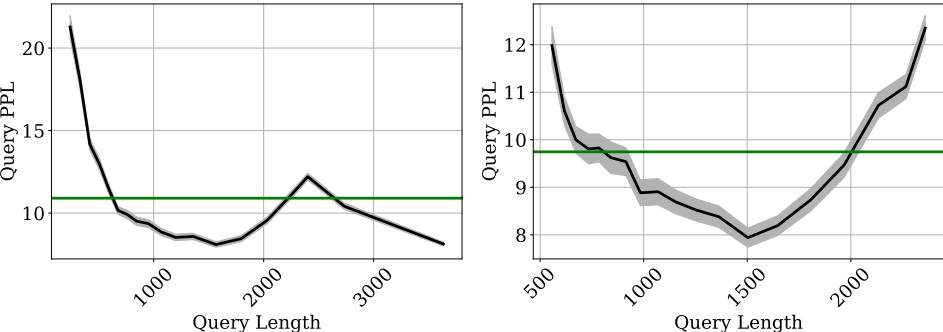

Figure 11: Due to the fact that the length of a text significantly impacts perplexity measurements, **(left)** we analyze the distribution of lengths present in the query set to identify a range of sequence lengths for which the performance measurement is relatively stable. **(right)** For the set of Open Orca questions, the best we are able to do is limit to between 500 and 2000 characters, where the query PPL varies according to a "U"-shaped and where most of the data are concentrated (the majority of the bins are located in the upper left of the left panel). Thus, the data is limited to this range for the figures presented in the main body to highlight whatever variance can be attributed to differences in density.

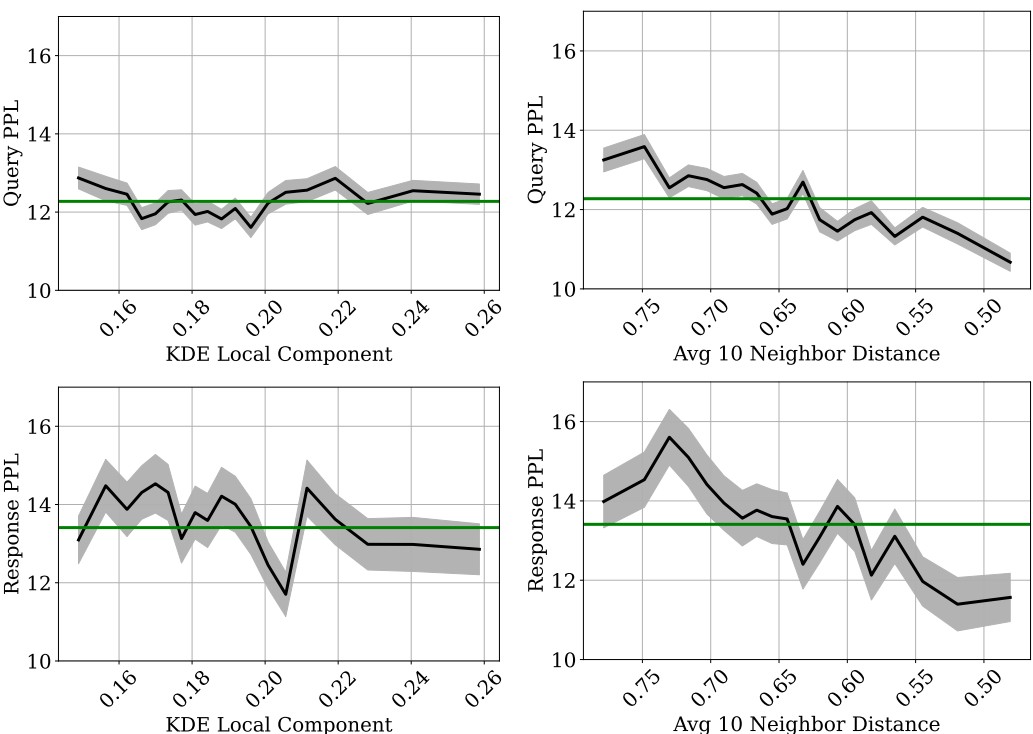

Figure 12: Perplexity according to Pythia 6.9B for questions from the MMLU test set (OoD) as a function of KDE with gaussian kernel and a bandwidth of 0.5 or average distance to k nearest neighbors, marginalized via equal mass binning into 20 bins. **Top,Left)** question perplexity vs the KDE with respect to only the local neighborhood within the corpus,**Top,Right)** question perplexity vs distance to k nearest neighbors, **Bottom,Left)** *response* perplexity vs the KDE with respect to only the local neighborhood, and **Bottom,Right)** response perplexity vs distance to k nearest neighbors. Horizontal line denotes the average across all queries. We see that while there isn't a consistent trend in question or response perplexity according to the local KDE, there is a more clear correlation for the distance to the k nearest neighbors.

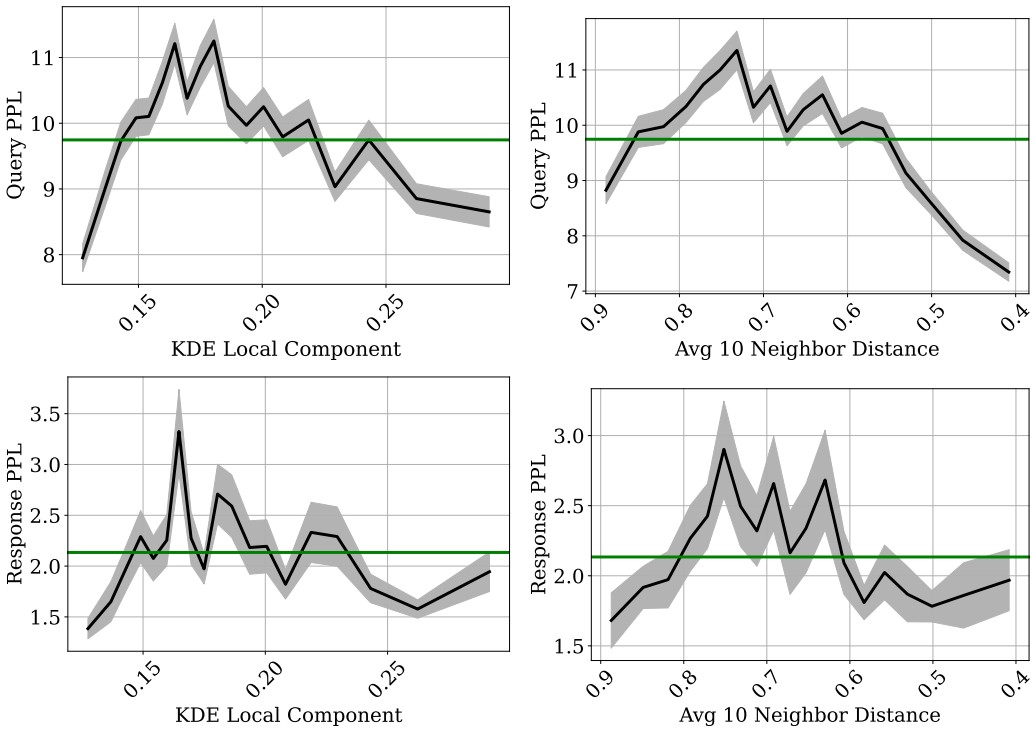

Figure 13: Perplexity according to Pythia 6.9B for questions from the Open Orca query set (OoD) as a function of KDE with gaussian kernel and a bandwidth of 0.5 or average distance to k nearest neighbors, marginalized via equal mass binning into 20 bins. **Top,Left)** question perplexity vs the KDE with respect to only the local neighborhood within the corpus,**Top,Right)** question perplexity vs distance to k nearest neighbors, **Bottom,Left)** *response* perplexity vs the KDE with respect to only the local neighborhood, and **Bottom,Right)** response perplexity vs distance to k nearest neighbors. Horizontal line denotes the average across all queries. We see that there is a non-monotonic relationship between query perplexity and both of the density heuristics. For response perplexity the trend is noisier. We suspect that the length variation for this query set washes out the effect of relative density differences.

## A.9 In-the-Wild Experiments: Full Regression Results

**Predicting perplexity with KDE computed w.r.t. The Deduplicated Pile for Pythia-6.9B (GLMER)**

$DV \sim KDE_{K,h=0.5} + (1|len(d_q))$

*Note: We use the "local" density estimate for regressions including density as a fixed effect, and report as ($\hat{\beta}$/p-value). "ns" denotes no significant effect.*

| Query Set | $KDE_{g,0.5}$ | Avg 10 NN |
|---|---|---|
| | $DV = $ Query Perplexity | |
| Rand. 10k (ID) | $\hat{\beta} = -62.250, p < .001$ | $\hat{\beta} = 49.388, p < .001$ |
| MMLU Test (OoD) | $ns$ | $\hat{\beta} = 8.5067, p < .001$ |
| Open Orca (OoD) | $\hat{\beta} = -6.9712, p < .001$ | $\hat{\beta} = 1.5845, p = .001$ |
| | $DV = $ Response Perplexity | |
| Rand. 10k (ID) | N/A | N/A |
| MMLU Test (OoD) | $\hat{\beta} = -53.911, p = .001$ | $\hat{\beta} = 16.737, p = .002$ |
| Open Orca (OoD) | $ns$ | $ns$ |

## A.10 Preliminary Investigations

### A.10.1 Paraphrase Retrieval

The LMD3 hypothesis relies on the fact that mixing exact copies of samples $x$, or paraphrases thereof, into the training corpus $X$ should increase the KDE at query point $x$. One sufficient condition for this to be true is that the paraphrases we mix into the pretraining dataset must be represented by the embedding model as *more similar* on average to their corresponding original query than the nearest neighbors of other training samples are on average.

In order to confirm that the lightweight embedding model we use is adequate for the similarity computations required by the KDE, and thereby limit the likelihood that the neighbor-search step would confound our overall results, we formulate a retrieval problem using the queries and their paraphrases and confirm that our nearest neighbor search reliably retrieves the "correct" neighbors for each query. In addition to that test, we visualize the distribution of the nearest neighbor distances for the $1,000$ queries where we plant 3 paraphrases and 1 exact copy, for those queries that we do not.

We see that due to the presence of the exact copy of each query for the interventional subset, in the left panel of Figure 14, for queries with leakage, the nearest neighbor is at distance 0.0, and due to the presence of the paraphrases, the average distance to the top-3 nearest neighbors is also much smaller than for queries with no paraphrases or copies. We corroborate this visualization by treating the collection of copies and paraphrases for each question as the target set in a retrieval problem and measuring Recall@k for $k \in \{5, 10\}$. While a perfect score of 1.0 is theoretically achievable at just $k = 4$ for this data, while neither model achieves this score even at $k = 5$, at $k = 10$ both are over 92% which we treat as sufficient evidence that the local neighborhoods of queries will contain the paraphrases and copies we introduce with some reliability. Thus, we conclude that the embedding space generated by the lightweight retrieval embedding model is unlikely to cause experiments to return null effect relationships with the KDE for the spurious reason that the embedding space doesn't return the "correct" neighbors.

### A.10.2 Bandwidths for separability of leak and non-leak

### A.10.3 Training to high accuracy on the test set.

|            | Retriever | Mistral 7B |
|------------|-----------|------------|
| Recall@5   | 0.9075    | 0.8875     |
| Recall@10  | 0.9375    | 0.925      |
| NDCG@10    | 0.939     | 0.928      |

Table 7: During preliminary explorations we set up a retrieval evaluation where we used the original test questions as queries, and tried to retrieve their paraphrases (3 per question plus 1 exact copy) when mixed into the training corpus. The embeddings produced by both the very strong LLM being analyzed for performance effects, and lightweight retrieval model, were equally useful in searching for leaked copies of the test questions when considering the 5, and 10 nearest neighbors (Recall@k) for each test question query. In light of the fact that we wanted to scale our experiments to pretraining corpora, we decided that it was acceptable to use the lightweight retrieval model for the main experiments.

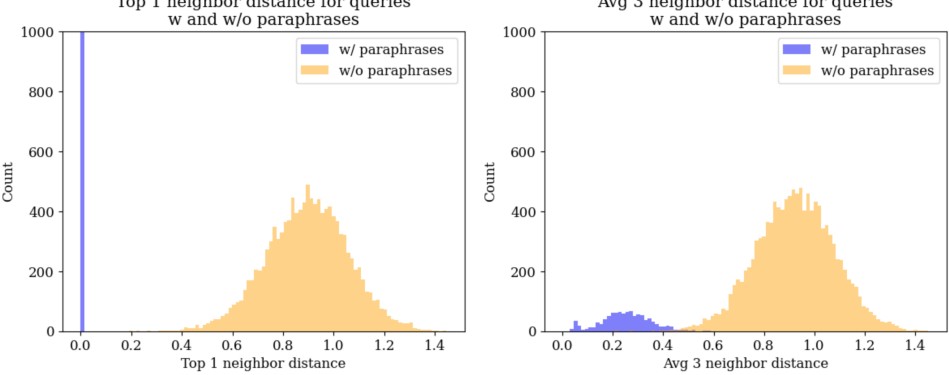

Figure 14

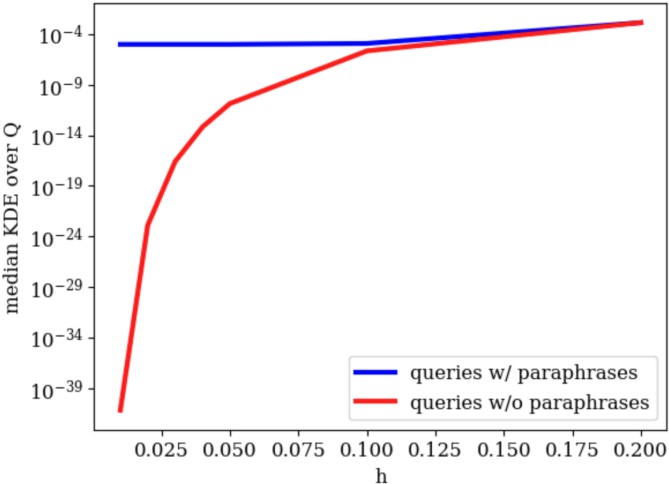

Figure 15: Since the proper KDE bandwidth for a particular problem an empirically derived parameter, we run a small ablation to chose a selection of bandwidths that are likely to produce an informative range of KDE values, especially those useful for identifying the set of queries for which we intervene through the planting of paraphrases amongst the other test questions. We see that as bandwidth is swept from low to higher values, the difference between the KDEs for queries with planted paraphrases and those without becomes smaller and smaller. This suggests that narrow bandwidths close to 0.0 are likely the most useful for developing a reliable estimator of whether a query has relevant paraphrases included in the training dataset. We use a representative selection of bandwidths from across this range in our main experiments so that we are able to examine what effect the bandwidth has on the relationship between KDE and other measures of interest. For the two euclidean kernels we investigate, the bandwidth selections are $\{0.01, 0.05, 0.1, 1.0\}$ for the exponential kernel, and $\{0.1, 0.2, 0.5, 1.0\}$ for the gaussian kernel, which we identified as reasonably similar sets for both kernel functions. We discuss the potential limitations of the selection process in the main work.

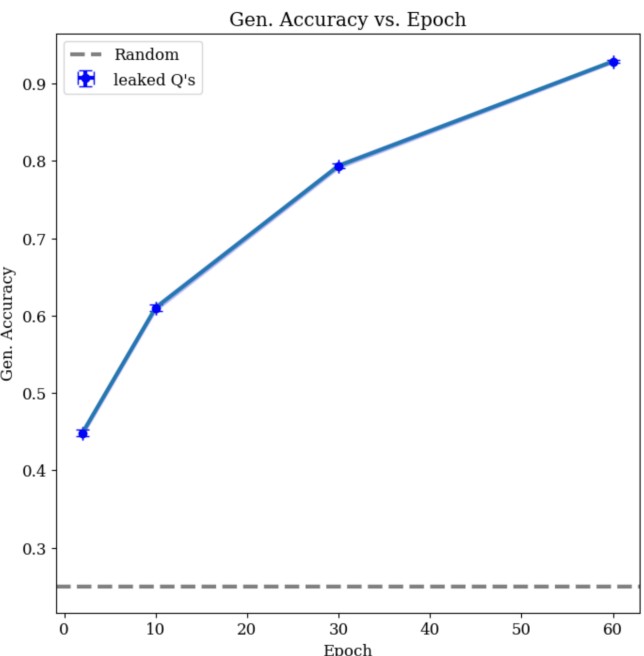

Figure 16: In order to ground our expectations on what performance we expect to see given the realistic finetuning hyperparameters we utilize, we perform a final validation check experiment by training for an extended number of epochs on a dataset that solely consists of the MMLU test queries. Mirroring the observations of Yang et al. (2023) we see that an extremely large number of epochs of training are required to even approach perfect performance on the test in the "ideal" scenario of training on the exact test queries. Since we both focus on the more realistic setting of training just a couple epochs, as well as on a larger set of training data where only a limited number of test queries and or their paraphrases are mixed in, much lower performance is expected even on the leaked test questions than the extreme values achieved beyond 30 epochs.

