# OpenReview forum: "LMD3: Language Model Data Density Dependence"
_colmweb.org/COLM/2024/Conference — COLM_

### Official Review · Reviewer_phfD · 2024-04-29

**Rating:** 6
**Confidence:** 4
**Ethics Flag:** 1

**Summary:**

The paper considers whether kernel density estimation can be used as a measure to quantify the familiarity (my term, not the paper's) of a test-time example to a language model.  The paper includes a "controlled" experiment where MMLU test instances and/or paraphrases of them are leaked into finetuning data (and effect on their accuracy is measured), and "in the wild" experiments where test instances are leaked into pretraining data (and effect on their perplexity is measured).

**Questions To Authors:**

Suggestion:  figure 2 is really nice.  I would suggest using the number of instances as the x-axis on the left, it's just easier to interpret and doesn't rely on weird assumptions (i.e., your fractional weighting based on cosine similarity).  The wide error bars on the right side make me want to see plots that show individual points (for a single choice of what is "leaked" at a time), for individual data points' KDE values and accuracy values.  Do we see correlation across instances as well?

**Reasons To Accept:**

At a high level, the method is intuitive and straightforward, and arguably a very strong starting point.

The findings suggest that KDE is a useful tool for this kind of analysis.

The paper describes its own applicability as a limitation; the pretraining corpus needs to be available to carry out the analysis on a model, in general.  I don't think this is a strike against; the more concretely the research community can explain why the data needs to be open, the stronger the pressure will be for transparency in how models are built.  This paper can be part of that case.

I found the paper generally clear.

**Reasons To Reject:**

One matter that's not at all clear from the paper is the matter of text granularity.  In the KDE setup, the paper doesn't make clear what an item in the corpus is.  Presumably it's not a word/token; is it a document?  Something smaller?  It would seem to make a big difference.  I'm fairly certain that this is a distinct decision from "bandwidth" and other kernel hyperparameters.

There is no discussion of the data that the off-the-shelf embedding model was trained on.  It seems worth some discussion as contamination of the embedding model's training data (with test query content) could be a problem.

It's a weakness of the paper that the Algorithm 1 and the details of paraphrase generation are deferred to the appendix.  I suspect other reviewers might prefer some other appendix materials to have been in the main paper, and I empathize with the challenge of keeping to the page limit.  But these two, at least, are essential for understanding the paper.  The details of the mixed effects models and controls for length would also be better in the main paper.  (Some space could be saved by compressing the related work section and figure 3's "count" histograms, could maybe be cut or moved to appendix since they're very low-information.)

I'd strongly recommend reorganizing the paper so that results of an experiment come directly after the explanation of the setup.  The current organization (all experimental setups laid out before discussing any results) creates an unnecessary mental burden on the reader.

Section 5.4:  MMLU test instances are taken as OOD, but the paper doesn't say whether these were explicitly checked against the pretraining corpus.  Perhaps some of them are actually in the pretraining data.

My understanding of the methodology is that "leaked instances" (in the pretraining case at least) are revealed to the model only after the normal pretraining is complete (the authors did not re-pretrain the whole model with the leaks randomly mixed in).  This has implications for the role of pretraining hyperparameters and the generalizability of the conclusions.  The paper should discuss this limitation.

There is no attempt to quantify the computational cost of the experiments, leading to uncertainty about how practical the method is to apply in practice or to experiment with further.

This claim in the introduction -- "the effect of dataset leakage is small enough that it likely does not invalidate benchmark numbers" -- is not really supported by the quantitative arguments of the paper.  The paper doesn't really inspect any benchmark numbers at all.

---

> ### Author Rebuttal · Authors · 2024-05-31
>
> (Responses in comment order, space permitting.)
>
> In Appendix A.1.1 we discuss segmentation in some depth and in Sec. 8, we remark that it is an overlooked detail in similar works. However, in Sec 5.4 in the main body, we now realize that there is no pointer to Appendix A.1.1, a mistake on our part.
>
> The off-the-shelf embedding model we use is detailed in Appendix A.2.1. We believe that this model was trained on a mixture of retrieval specific data including MS-MARCO. While we can not completely rule out the possibility of overlap between MMLU and those data without significant work, the likelihood that the retrieval specific training data for that model contains any significant fraction of MMLU test question texts is exceedingly unlikely in our opinion.
>
> In response to this comment and the suggestions of other reviewers, we are choosing to remove the histograms from Fig. 3 in the main body. Please see the [attached](https://anonymous.4open.science/r/colm-rebuttal-8CB1/LMD3_colm_rebuttal.pdf) set of updated figures for mockup of the updated version. In addition, Algorithm 1 will be promoted to the main body using the extra page of space allowed in the updated draft.
>
> We appreciate the suggestion for improved clarity via a "Setup-Result-Setup-Result" ordering and will consider it in the draft update process.
>
> While we cannot entirely rule overlap out, none of our results or claims hinge on "OoD" separation between MMLU and the Pile being strict. Rather, in the pretraining scale experiments we report relative differences where the training corpus and the model are both constant.
>
> We would be happy to add something like the following in the limitations section:
>
> "Conducting a similar series of controlled experiments where examples of test queries are leaked into a full scale pretraining dataset would be interesting but was computationally out of scope for this work. We expect that results could vary and that the KDE hyperparameters explored in this work might not be optimal for that setting."
>
> Regarding algorithmic details and computational cost, see response to reviewer vEjv.
>
> Agreed, we will tighten the specificity of the claim about benchmark invalidation in the introduction or omit it altogether.
>
> We thank the reviewer for the insightful suggestion and have prepared a version of Fig. 2 where the x-axis represents "counts" i.e. if we train for 2 epochs, and have leaked 1 exact copy and 1 paraphrase, the total count would be $2*(1 + 1) = 4$.

---

> > ### Comment · Reviewer_phfD · 2024-06-04
> >
> > Thanks for the response; acknowledged.

---

### Official Review · Reviewer_y6ai · 2024-05-14

**Rating:** 4
**Confidence:** 4
**Ethics Flag:** 1

**Summary:**

This work studies the data density dependence of language models on the perplexities/accuracy of queries / downstream tasks. In particular, this work provides the framework to assess the dependence of performance on a query given support of the training distribution The work suggests that model performance can be reliably predicted based on deter density dependence in many of the cases.

**Questions To Authors:**

1. here was a recent paper listed as no zero shot performance without exponential data (https://arxiv.org/abs/2404.04125). I wonder if the authors would like to contrast the results with the same. of course this paper came out during the review. But I am curious if you think exponential amount of data should be added rather than the linear amounts that tried in your work.
2. secondly I don't understand the goal of figure 3.a, the count figure.
3. When the neighbor distance is low, do you check if there is a paraphrase or exact duplicate? Since you're only aggregating with 10 nearest Neighbors, we can actually check if the samples that were in the train set were in some way duplicates of the query
4. How does the average cosine similarity fare in comparison to the distance between embeddings

**Reasons To Accept:**

The main strength of this work includes a very comprehensive study of different models across various tasks.

I like how the authors performed experiments to understand how the KDE changes as duplication increases. The control experiment by purposefully adding paraphrases of test set queries to measure the effect of the same on model performance helps understand the effect of data density on model performance.

Some of the takeaways of this work are quite interesting. For example, model performance does not increase drastically even if exact duplicates are added to the data set. And for new tasks, you can give some estimate of model performance based on training density.

**Reasons To Reject:**

1. Embedding model bias. I would like to hear more from authors on this beyond the small subsection of the paper. For example, embedding models not trained on code will actually cluster all code data very close by. I think the embedding model bias is a big problem here.
2. I do not understand why should we use KDE, when avg nearest neighbour distance performs better in the last few graphs in the paper? I did not find this discussion in the paper.

On top of that, two weaknesses that question the validity of the hypothesis of this paper:
1. The paper does not discuss the result of adding only Paras=3 in Figure 2. This experiment should not increase the density more than Paras=1 2 3 because the point in yellow should have almost 3x more density as compared to Paras=3. However, if it results in close to equivalent improvement in model performance, it would suggest density dependence may not actually be the true hypothesis here. Some other tests would be, does repeating exact 3x help more than paras=1,2,3 + exact repetition? In general, the "exact" inclusion convulates the hypothesis quite a bit. One example, should not increase the density much, but it does have a very large impact on the model performance (as compared to Paras=1,2,3)
2. In Figure 5, we never discuss accuracy on MMLU and only the response and query PPL. This is not really descriptive enough because as models improve or go to more favorable regions, the perplexity typically improves on all correct and wrong answers. I do not think the graph is suggestive of model performance as of now.

---

> ### Author Rebuttal · Authors · 2024-05-31
>
> (Responses in comment order, space permitting.)
>
> Beyond questions about an embedding model's capacity to represent certain data and questions about the diversity of its training data, we remark that interpreting distances in high dimensional spaces can be difficult. However, we would like to present a preliminary analysis that didn't make it into the review copy. In early experiments we considered embeddings derived from the hidden states of the LLM you are analyzing as an alternate to the standalone lightweight retrieval encoder used throughout the work. See the [attached](https://anonymous.4open.science/r/colm-rebuttal-8CB1/LMD3_colm_rebuttal.pdf) document for this analysis; we will incorporate it into the updated draft.
>
> Regarding the differences between KDE and top-k distance aggregations please see the response to reviewer vEjv.
>
> First we note that due to computational limits, and the cost of preliminary explorations, we chose not to run all possible permutations of the finetuning experiment focusing on those determined to be most critical to exploring our hypothesis.
>
> We have updated Fig. 2 slightly in the attached document for improved clarity. While it's intuitive that training on an exact copy of a test question causes a larger increase in model performance on the corresponding questions than training on a single paraphrase, the fact that it has a larger effect on accuracy than training on all three paraphrases is slightly surprising. We adopt a purely descriptive stance in interpreting these results.
>
> In the pretraining scale experiments we focused on loss/perplexity as these measures can be calculated on any text sample rather than accuracy measures that are constrained to only task-like data.
>
> This (2404.04125) is a very interesting work, thank you for pointing it out! While it is difficult to identify a precise correspondence between visual concept frequencies and the repetition of test questions or paraphrases thereof, we agree that the there is some relationship between these findings.
>
> While we cannot recover the precise segement texts for each neighbor easily at this time, we can report some anecdotal observations that nearest neighbors for pretraining segments were indeed sometimes duplicates or outher chunks of the same document.
>
> While we did not directly compare the difference between cosine similarity and unnormalized distances, the sentence-transformers embedding model we used emits unit normalized vectors by default.

---

> > ### Author Response · Authors · 2024-06-06
> > **Discussion period closing soon**
> >
> > Hello again and thank you for the time already invested in your review of our work!
> >
> > As the discussion period is coming to a close today, the authors are wondering whether the reviewer has had a chance to look through our rebuttal and linked figure updates? We'd be happy to try and address any specific lingering concerns from the reviewer that we weren't quite able to treat properly under the tight rebuttal length constraint.

---

> > > ### Comment · Reviewer_y6ai · 2024-06-06
> > >
> > > Thanks for your efforts with the rebuttal response. I do not think my questions are satisfactorily answered, and my main qualms still remain. In order of priority, they are
> > >
> > > 1. When nearest neighbors work just fine (except for the value of k being brittle?), I feel this paper is trying to make a claim of data dependence that is not really the causal reason for improved performance. Two things that could help in the future: (i) Do an actual ablation showing how duplicates change the density versus how they impact the nearest neighbours score. (ii) Actually show how brittle NN is. In fact, answer to Q3 above further suggests the existence of duplicates.
> > >
> > > 2. The reason why I suggested the experiment Paras=3 experiment is not just to pad the paper with more experiments, but because I believe this is a critical experiment in understanding the cause behind increase in model performance.
> > >
> > > 3. The MMLU comment has reason: Often when models become better, their PPL on wrong and right answers go low together. To understand this better, you can do the following : evaluate MMLU performance, but this time by checking PPL on second most likely (but wrong answer). You will likely see the exact same trend.

---

> ### Author Response · Authors · 2024-06-06
>
> Thank you for your continued engagement in the review process!
>
> First, we would like to remark on the close relationship between duplication and density, as it seems to be a relevant distinction to discuss. In the review copy (Section 2.1 sentence 2) we note that "Repetition is an important extremal case of elevated density corresponding to a spike in the density function at a specific point in the sample space" and as such we expected duplication to have a strong effect. By design, we included it in our controlled experiments. We also agree that it is relevant to our in the wild experiments. However, we would like to underscore the fact that the presence or lack thereof of duplicates, which are expected to affect both density measures and top-k distance measures, does not invalidate any particular experiment. It also would not invalidate any particular claim about density as duplication _is_ an instance of increased density, albeit an extreme one.
>
> Additionally, regarding para=3, we chose to include paras=1, paras=12, and paras=123 in order to make the results cleaner and easier to understand. The cumulative relationship between the chosen paraphrase settings means that density would be expected to grow across those three levels of leakage in an easy to interpret manner.
>
> Next, in light of recent comments 1. and 2. the authors took a step back and considered an alternate modeling setup, one where we directly ask "can the two features, density and top-k distance, can explain _different_ aspects of variance?" We realized that this could be achieved by running an particular regression model that asks whether the two features explain significant parts of the variance in the performance outcome, when controlling for the variance explainable by the other.
>
> ```
> loss_rank_acc ~ gaussian_0.1 + avg_10_neighbor_distance + (1 | query_length) + (1 | task/idx)
> ```
>
> Our predictors ("Fixed effects") are `gaussian_0.1` and `avg_10_neighbor_distance` and target is `loss_rank_acc`. The other features of `query_length` and the subtask id from the MMLU subject list and the particular question id (i.e. `task/idx`) are treated as "Random effects" that the model attempts to marginalize out.
>
> ```
> Fixed effects:
>                            Estimate Std. Error  z value Pr(>|z|)
> (Intercept)               1.416e+00  3.514e-01    4.029 5.60e-05 ***
> gaussian_0.1              4.625e+04  5.372e+00 8609.363  < 2e-16 ***
> avg_10_neighbor_distance -3.595e-01  8.711e-02   -4.127 3.68e-05 ***
> ---
> Signif. codes:  0 ‘***’ 0.001 ‘**’ 0.01 ‘*’ 0.05 ‘.’ 0.1 ‘ ’ 1
> ```
>
> We want to look at the `Estimate` of the coefficient, and the `Pr(>|z|)` or p-value representing the significance of the predictive relationship for each feature. Interpreting the coefficients, we observe that as density goes up, rank accuracy goes up ($+$ coef) and as neighbor distance goes up rank accuracy goes down ($-$ coef) as expected. Next, moving to the p-values, the model suggests these relationships are accounting for _unique variance_, i.e., the small p-values indicate that both terms remain reliable predictors of variance when included in the same model (*** level or better).
>
> This evidence clearly shows that despite the intuitive and theoretical ways in which top-k distances and the kernel density measure are related, in practice, they can each be used to explain different parts of model behavior. We agree with the reviewer that studying this relationship further is an important area for future work. Overall, we are very thankful to the reviewer for prompting us to undertake this analysis! We will perform a more complete set of similar regressions for the camera ready copy that helps the work more clearly communincate the difference between our proposed methodology and the top-k distance baseline.
>
> Finally, regarding 3. we agree that the relationship between multiple choice task performance and the likelihoods assigned to individual responses is complicated. That said, the experiment that the reviewer proposes has roughly two outcomes. Either, the same trend in PPL on the correct answer as a function of density or neighbor distance is observed when computing PPL on incorrect answers, or it is not/the trend is different. This experiment would not really build any extra evidence about the overall hypothesis regarding performance and training density. In particular, finding that a trend or significant predictive relationship exists in both evaluation setups (MMLU+correct answer and MMLU+incorrect answer), would actually only suggest to you that PPL is probably a weak proxy for discontinuous evaluations like MCQ accuracy. However, this is a known result in the literature, and the models we had access to are too weak to solve MMLU well in the first place. When it was possible to do so, in our controlled experiments, we focused on the more meaningful task accuracy measure rather than PPL.

---

### Official Review · Reviewer_vEjv · 2024-05-17

**Rating:** 5
**Confidence:** 4
**Ethics Flag:** 1

**Summary:**

The authors present a series of studies relating aspects of training data density at the example level to perplexity performance and task performance in language models. The authors perform both 1) a controlled experiment in which they vary the number and type of examples in training data that are "leaked" from the test set, and 2) an analysis of the relationship between perplexity measures and two different estimates of data density. Their claims are that increasing the training data density around certain examples in the test set can boost task performance, and that measures of training data density around test set examples can be predictive of perplexity performance.

**Questions To Authors:**

See above. Is KDE truly not affected by the existence of "leaked" paraphrases? Additionally, some minor comments/suggestions/questions:
1. Last sentence in section 4.1 does not parse well to me. I would recommend not only splitting it up into multiple sentence but also rearranging the post-comma clause (e.g. --> "Future work can explore ...")
2. The term "doctored" in 5.1 makes that sentence read weirdly to me. Is there a more specific way that can the modifications to the training set can be succinctly described there?
3. It would be helpful to point to 6.1 in Figure 2 for a definition of the "effective epochs" that the accuracy values are plotted against, since 6.1 falls on a later page.
4. Figure 3 caption: "exactly and **/** or via paraphrase"
5. Figure 5 caption: would like to see a pointer to the missing plots in the Appendix
6. Appendix 10.2 and 10.3 seem to contain only figures; unclear which specific figures are meant to belong where

**Reasons To Accept:**

1. The experimental design and methodology are both sound and grounded strongly in realistic settings.
2. The experiments themselves and the research questions they are meant to answer are interesting and relevant to many researchers who work on or with language models
3. Mostly clear writing and organization

**Reasons To Reject:**

My main concern is that at least one of the main claims about the usefulness of kernel density estimation (KDE) seems only weakly supported; it is unclear to me whether my perception is due to the evidence gathered itself or simply the authors' presentation of the results.
1. The authors claim that Figure 3 "shows that KDE is a discriminate feature between the leak set and non-leak set." However, the top row of Figure 3 clearly shows overlapping bars in what appears to be a histogram of KDE values. If there is any distinction, it is not visible from that plot with the x-axis scale presented. (Please do let me know if there is some mistake in Figure 3 or if a different scale would show a different pattern.) There appears to be a clear distinction only when the "leaked" queries are exact duplicates of test queries, which significantly dampens any claims of KDE's usefulness as a "discriminative feature" between leaked and non-leaked data points (if it cannot detect "leaked" paraphrases + many other simple methods could detect exact match leakage)
2. Relatedly, Figure 5's caption and 6.2 show that KDE is often but *not consistently* indicative of predictive of perplexity (visible also in Appendix figures). I would like to see the authors discuss more specifically the usefulness and limitations of KDE and top-k neighbor distance -- especially since it seems like the average Top-k neighbor distance may be at least as reliable and KDE seems to feature more heavily in the manuscript

Other concerns:
3. No useful information about the proprietary algorithm for example neighbor search. The efficiency of this implementation seems to be an important aspect of the authors' methodology, so it could be very helpful to mention its runtime complexity or even the scale of resources (compute/time) required to run the analyses presented
4. Baseline rank accuracy (0 paraphrases, Exact=0) missing depiction of variance. I would have liked to see it plotted like the others with error bars (or bands if the horizontal line style is kept)


I would not be comfortable accepting this paper in its current state, but I would be sincerely glad to adjust my score if the authors address my concerns, as I find the topic and experimental setting interesting, and my concerns are more about the discussion of the results and the conclusions drawn than the fundamental questions that the authors set out to answer.

---

> ### Author Rebuttal · Authors · 2024-05-31
>
> (Responses in comment order, space permitting.)
>
> In the [attached](https://anonymous.4open.science/r/colm-rebuttal-8CB1/LMD3_colm_rebuttal.pdf) set of figures we have mocked up some requested changes, a few of which we discuss below.
>
> We agree that the first row of subplots in Fig. 3 as presented in the review copy has relatively low information content. Therefore, we propose removing the histograms entirely from Fig. 3 in the main body, and then in the Appendix, presenting a "zoomed in" version of the histograms for the row of cases where only paraphrases were leaked.
>
> As the reviwer points out, our experiments suggest that the simple average of the distances between a query and its $k$ nearest neighbors is _also_ a discriminative feature. To be incorporated in the updated draft:
>
> "The KDE methodology stands apart from simpler aggregations like the top-$k$ distance measure in its "smoothness". The top-$k$ average requires that you choose a specfic value of $k$ and neighbors that lie beyond this boundary will not be accounted for. Changing $k$ by just a value of $1$, can drastically change the mean. In contrast, the KDE measure accounts for the nearest and farthest neighbors simultaneously and varies more smoothly under perturbations of the $k$ nearest neighbor set. While one does still need to choose a kernel bandwidth for the KDE, informally, there always exists some $\delta$ such that the change in density at query point $q$ for bandwidth $h$ versus bandwidth $h+\delta$ is bounded and small."
>
> Overall, we believe that choosing k, which can have a discontinuous effect on the estimate, is more fraught than tuning the bandwidth and it is a limitation of our evaluation rather than the method that we didn't identify a scenario where the KDE and top-k distance approaches diverge in discriminative power.
>
> We apologize for the lack of detail. For the larger scale nearest neighbor searches over pretraining scale datasets, we used a system called ScaNN (Scalable Nearest Neighbors, (Guo, 2020)). In line with this request and comments made in another review, we will promote Algorithm 1 to the main body in the updated draft. Additionally, we can add a discussion to accompany the algorithm:
>
> "Using a brute force maximum inner product search, the time complexity of the neighbor search step for $M$ queries over a corpus of $N$ segments is $O(M*N)$. However, with more clever tree data structures and quantization schemes, this runtime can be reduced significantly."

---

> > ### Author Response · Authors · 2024-06-06
> > **Discussion period closing soon**
> >
> > Hello again and thank you for the time already invested in your review of our work!
> >
> > As the discussion period is coming to a close today, the authors are wondering whether the reviewer has had a chance to look through our rebuttal and linked figure updates? We'd be happy to try and address any specific lingering concerns from the reviewer that we weren't quite able to treat properly under the tight rebuttal length constraint.

---

> ### Comment · Reviewer_vEjv · 2024-06-06
>
> Thank you for your response! I disagree with the decision to remove the histograms entirely. Imo the zoomed in version is fine and informative, as long as you are clear about the differing x-scales.
>
> I appreciate the discussion of sensitivity to k. However, it is not immediately obvious to me why it would be preferable to tune the KDE bandwidth than the k. "There always exists some $\delta$ such that the change in density at query point $q$ for bandwidth $h$ versus bandwidth $h+\delta$ is bounded and small" -- is it also true that it is easy to find such delta? Are optimal values for one hyperparameter expected to hold more stable in across settings?
>
> Also, fwiw I agree with reviewer y6ai's assessment that it would be extremely informative to see isolated Paras=3 in Figure 2, given that 1, 2, and 3 are sorted in "ascending order by similarity." I would appreciate some sort of verbal commitment to running this ablation before camera ready.
>
> I am willing to slightly increase my rating, but I believe there is room for improvement before publication (Edit: increased rating from 4->5)

---

> > ### Author Response · Authors · 2024-06-06
> >
> > Thank you for the response and the increased rating!
> >
> > Regarding the histograms in Figure 3, we're glad that you think the zoomed in version is informative. If there is no conflicting request from the other reviewers, then we can certainly include a final version of Figure 3 that swaps in the zoomed in histograms for the paraphrase only cases, and add a clear description of the axes scaling for that row in the caption.
> >
> > Regarding the KDE bandwidth $h$ versus the value of $k$, we'd like to clarify that the rebuttal comment about the "$h+\delta$" is not about actually finding such a $\delta$, but rather a comment about the _smoothness_ of density wrt the users choice of $h$, versus the lack of smoothness of top-k wrt the users choice of $k$ (imagine one chose $h$ to be slightly off from "optimal" versus $k$ that is off by 1 or more).
> >
> > That said, we'd like to step back and remark that this is actually a classic choice in general for nearest neighbor data structures. Thinking simply of the nearest neighbor search process abstractly, one can either retrieve the "nearest k points" in the space, or can retrieve the "points that are closer than some radius d". A KDE measure with some bandwidth $h$ is actually a relaxation of the latter, as it is effectively building evidence at a query point based on a _soft_ analogue to a radius.
> >
> > Relevant to this conversation, we also think that the analysis presented in the recent response to y6ai is quite informative here. We find that since the KDE and the top-k measure seem to account for _different_ aspects of the variance in our setting, that there exists complementarity between the KDE and the top-k measures. Using the extra page of space, we think a concise presentation of this analysis and a discussion of complementarity of the two apporaches would make a good addition towards the end of the camera ready main body.
> >
> > Finally, in light of the discussion with y6ai and your most recent comments, between now and camera ready, we are happy to agree to run the requested training experiment with just the singular nearest paraphrase (para 3), and analyze it in the same manner as the other settings. This would mean visualizing it with the other settings within Fig. 2, and in Fig. 3.

---

### Official Review · Reviewer_tSwb · 2024-05-21

**Rating:** 7
**Confidence:** 4
**Ethics Flag:** 1

**Summary:**

This paper proposes a methodology for establishing a relationship between LM task performance and training data density estimation. Some controlled experiments are performed by introducing paraphrased data points. Although the approach is interesting, the results are not always easy to interpret. Nevertheless, I think the paper provides a useful tool for analysis. Many details are provided in the Appendices.

**Reasons To Accept:**

The main reason to accept the paper is the analysis provided in the paper. I believe more analysis papers are needed in this field.

**Reasons To Reject:**

As the authors also point out there are some limitations which might restrict the wide application of the methodology.

---

> ### Author Rebuttal · Authors · 2024-05-31
>
> We appreciate the vote of confidence from the reviewer in both the quality of our analysis and the completeness of our presentation.

---

> > ### Comment · Reviewer_tSwb · 2024-06-06
> >
> > Thanks.

---

### Decision · Program_Chairs · 2024-07-10

**Decision:**

Accept

**Comment:**

Although the reviewers make arguments for and against this paper, including some unresolved questions (possibly due to time as they were posted at the end of the rebuttal period). This is an important area and everyone agree the experiments are well designed and study an important class of problems. The relationships between data and end task performance is a hard one to figure out, and this papers add quite a bit to the discussion even if it doesn't have perfect answers on all fronts.